# Advancing food security: Rice yield estimation framework using time-series satellite data & machine learning

Varun Tiwari[1]*, Kelly Thorp[2], Mirela G. Tulbure[1], Joshua Gray[1], Mohammad Kamruzzaman[3], Timothy J. Krupnik[4], A. Sankarasubramanian[6], Marcelo Ardon[5]

1 Center for Geospatial Analytics, North Carolina State University (NCSU), Raleigh, NC, United States of America, 2 United States Department of Agriculture (USDA), Agricultural Research Service (ARS), Grassland Soil and Water Research Laboratory, Temple, Texas, United States of America, 3 Farm Machinery and Postharvest Technology Division, Bangladesh Rice Research Institute, Gazipur, Bangladesh, 4 International Maize and Wheat Improvement Center (CIMMYT), Dhaka, Bangladesh, 5 College of Natural Resources, North Carolina State University (NCSU), Raleigh, NC, United States of America, 6 Civil and Environmental Engineering, North Carolina State University (NCSU), Raleigh, NC, United States of America

* v.tiwari@ncsu.edu

**Data Availability Statement:** All relevant data are within the manuscript and its Supporting Information files.

## Abstract

Timely and accurately estimating rice yields is crucial for supporting food security management, agricultural policy development, and climate change adaptation in rice-producing countries such as Bangladesh. To address this need, this study introduced a workflow to enable timely and precise rice yield estimation at a sub-district scale (1,000-meter spatial resolution). However, a significant gap exists in the application of remote sensing methods for government-reported rice yield estimation for food security management at high spatial resolution. Current methods are limited to specific regions and primarily used for research, lacking integration into national reporting systems. Additionally, there is no consistent yearly *boro* rice yield map at a sub-district scale, hindering localized agricultural decision-making. This workflow leveraged MODIS and annual district-level yield data to train a random forest model for estimating *boro* rice yields at a 1,000-meter resolution from 2002 to 2021. The results revealed a mean percentage root mean square error (RMSE) of 8.07% and 12.96% when validation was conducted using reported district yields and crop-cut yield data, respectively. Additionally, the estimated yield of *boro* rice varies with an uncertainty range between 0.40 and 0.45 tons per hectare across Bangladesh. Furthermore, a trend analysis was performed on the estimated *boro* rice yield data from 2002 to 2021 using the modified Mann-Kendall trend test with a 95% confidence interval (p < 0.05). In Bangladesh, 23% of the rice area exhibits an increasing trend in *boro* rice yield, 0.11% shows a decreasing trend, and 76.51% of the area demonstrates no trend in rice yield. Given that this is the first attempt to estimate *boro* rice yield at 1,000-meter spatial resolution over two decades in Bangladesh, the estimated mid-season *boro* rice yield estimates are scalable across space and time, offering significant potential for strengthening food security management in Bangladesh. Furthermore, the proposed workflow can be easily applied to estimate rice yields in other regions worldwide.

**Funding:** The authors received no specific funding for this work.

## 1. Introduction

Rice is the staple food for more than half of the world's population, constituting 20% of the world's food supply [1]. By 2050, the world's population is expected to exceed 9.8 billion, and a 60% increase in food consumption is anticipated [2]. This growing global demand for food is underscored by the fact that 90% of the world's rice is cultivated in developing nations such as India, China, and Bangladesh [3], emphasizing its critical role in ensuring both regional and global food security [4].

Rice production in Bangladesh holds a critical place in the country's agricultural and economic landscape, with more than 165 million people in Bangladesh dependent on rice for their livelihoods, calorie intake, and food security [5, 6]. Furthermore, Bangladesh is a major exporter of rice [7], which accounts for 70% of the agricultural Gross Domestic Product (GDP) and one-sixth of the national income in Bangladesh [8].

In recent years, Bangladesh has experienced adverse effects on rice yields due to climate change, characterized by increased extreme weather events, such as droughts and floods during critical phases of rice cultivation [9]. This impact is attributed to the strong dependence of rice yields on weather conditions throughout the cultivation stages [10]. Moreover, Bangladesh's high vulnerability to climate change, reflected in its seventh-place ranking on the Global Climate Risk Index, raises significant concerns, especially in agriculture and rice production [11]. Moreover, Sarker et al., predicted that between 2005 and 2050, rice production in Bangladesh will decline by an average of 7.4% every year due to climate change [12]. Likewise, the National Adaptation Plan of Bangladesh (2030–2050) published by the Ministry of Environment, Forest and Climate Change (MEFCC), Bangladesh [13], along with the findings from Kaur el al., highlighted the looming threat posed to this crucial agricultural sector, particularly rice, by the escalating impacts of climate change [14]. Ghose et al. (2021) recently revealed that an increase in temperature by 1˚C and rainfall by 1% results in a decrease in *aman* rice yields by 33.59% and 3.37%, respectively [15].

The country's vulnerability to these climatic challenges poses a potential threat to food security. It heightens Bangladesh's susceptibility, impacting livelihoods, economic stability, and food security at both national and global levels. This will also pose a significant obstacle to the United Nations Sustainable Development Goals (SDGs), prioritizing achieving zero hunger and promoting sustainable agriculture by 2030 [12]. To manage food security, it is therefore important to accurately estimate rice yields, production and trends, which could also significantly contribute to efforts to lessen climate-related risks to rice production in Bangladesh. Hence, accurate and timely statistics on rice yields, and long-term trends could assist the government of Bangladesh in managing food security and contribute to efforts to lessen climate-related risks to rice production in Bangladesh.

Globally, efforts to quantify rice yield have traditionally focused on field-based sampling using crop cut surveys, which provide accurate yield estimates [13, 16, 17]. Such field-based sampling techniques are extensively used by government agencies in Bangladesh to estimate rice yields for national-level reporting [16, 18]. This involves a systematic selection of representative rice fields, physical harvesting of rice samples, and measurement and documentation of grain weights, followed by the extrapolation of these data to broader areas using statistical methods. However, these techniques cannot provide wide spatial and temporal coverage, and they are labor-intensive, time-consuming, and subject to sampling bias [19]. As a result, these methods cause delays in estimating rice yield and impede timely food management decision-making.

Conversely, remote sensing datasets, either optical or synthetic aperture radar (SAR), have shown their potential to quantify rice yield globally [20–22]. These datasets employ crop yield

data based on ground surveys and leverage dynamic crop growth simulation [23, 24], regression [25, 26], machine learning (ML) [27–29], and deep learning models to estimate rice yield [30, 31]. Moreover, these models make use of satellite-derived vegetation indices, including the normalized difference vegetation index (NDVI) [32, 33], enhanced vegetation index (EVI) [30, 34], soil-adjusted vegetation index (SAVI) [35]. Additionally, they incorporate meteorological data (rainfall and land surface temperature), soil information, and crop-specific parameters to estimate rice yields before the harvesting stage [36].

Similar remote sensing methods are utilized in Bangladesh to estimate rice yield, employing spectral bands and vegetation indices from optical and SAR images [37–39]. These methods also include regression-based approaches [40], ML models [41], and deep learning (DL) models [42]. However, these methods are primarily utilized for research purposes and need to be integrated into official reporting procedures. In addition, the rice yield maps developed from these studies need to be updated, and their coverage is limited to particular geographic regions within the country. Moreover, the rice yield estimation is often conducted at a coarse scale, typically encompassing districts, divisions, or the entire nation, thus needing more spatial and temporal precision. Currently, annual *boro* rice yield statistics are available from inventory data at the district level; however, no consistent yearly boro rice yield map is available for Bangladesh at a subdistrict spatial scale, such as a 1,000-meter spatial resolution. Due to this limitation, rice trend analyses conducted in other studies [9, 43] are typically done at district and national scales, rendering them less useful for planning and implementing policies at the micro-scale (sub-district level).

This study aimed to evaluate the potential of MODIS data and a random forest (RF) ML-based method to develop a rice yield model for estimating *boro* rice production in Bangladesh from 2021 to 2022 at a spatial resolution of 1,000 meters. Specific objectives were to 1) evaluate RF models for estimating rice yield in Bangladesh, 2) report the accuracy and uncertainty of *boro* rice yield estimation by RF models, and 3) understand the spatial and temporal trends of rice yield in Bangladesh from 2002 to 2021.

## 2. Methods

### 2.1 Study area

Bangladesh, a tropical nation with a land area of 148,460 km$^2$, is located in south Asia between the latitudes of 20˚44′00″ and 26˚37′51″N and the longitudes of 88˚0′14″ and 92˚40′08″E (see Fig 1A–1C, [44]). The dominant land use in the country is agriculture (65%), followed by forests (18%), urban areas (8%), and water bodies (6%). Bangladesh has 64 districts (Fig 1B) within eight administrative zones [45]. In addition, the agricultural areas in the county are classified into fourteen crop zones by the Bangladesh Agriculture Research Council (BARC, Fig 1B). The majority of land in Bangladesh is flat (with elevations below 10 meters above mean sea level), except for the Chittagong Hill Tracts in the southeast, where the average elevation ranges between 300 and 1,063 meters. Bangladesh has fertile agricultural land due to three major rivers—Ganga, Brahmaputra, and Meghna—which cater to the country's water needs.

Bangladesh has a tropical monsoon climate favorable for rice cultivation, and 75% of its agricultural activities involve rice production, which plays a significant role in the country's agricultural landscape. The country experiences three distinct rice-growing seasons in a year: winter (*boro*, planted from November to February), summer (*aus*, planted from March to May), and monsoon (*aman*, planted from June to October). The dominance of rice cultivation makes Bangladesh the world's third-largest rice producer [46, 47].

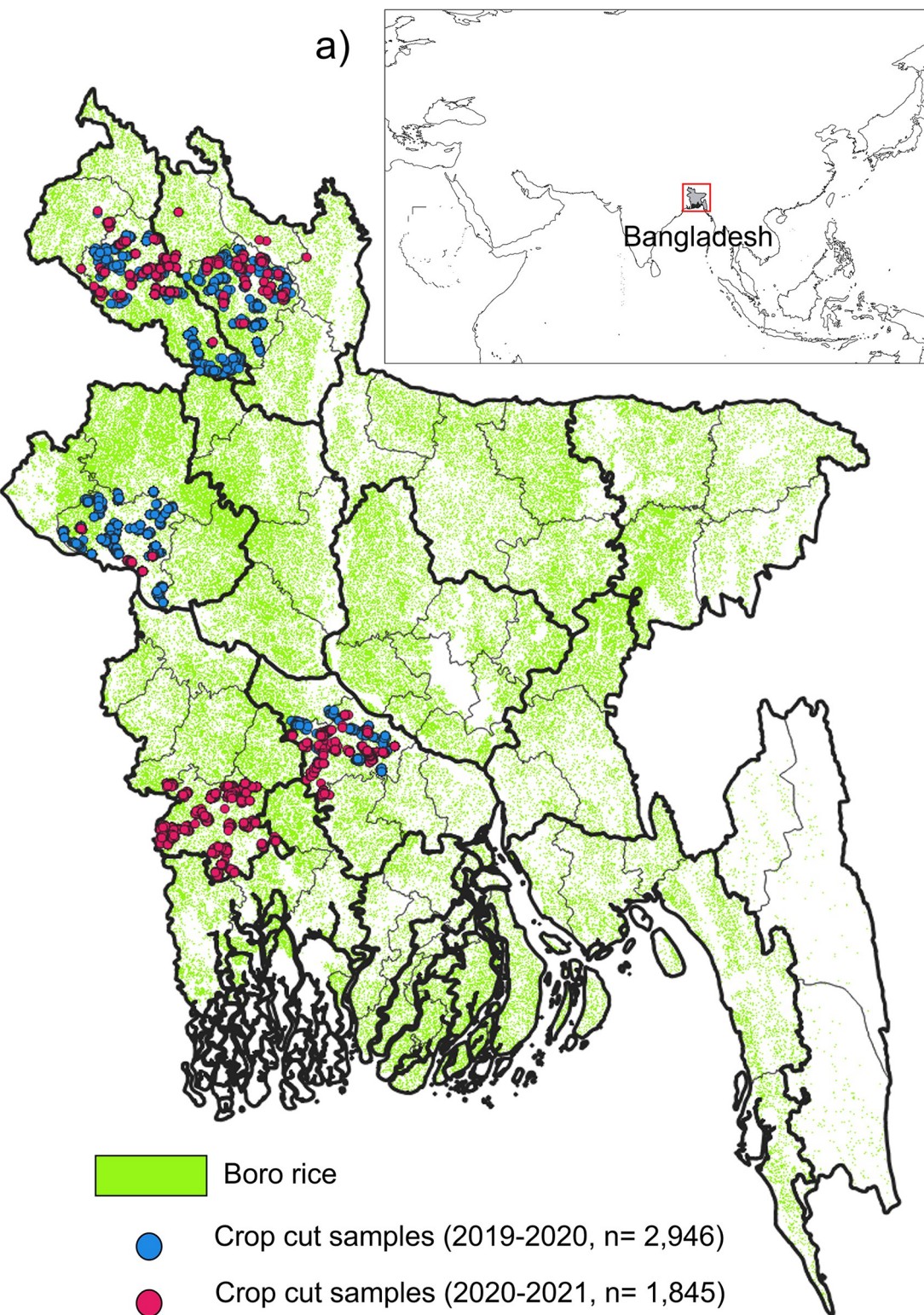

**Fig 1.** Map of the study area in Bangladesh showing a) the location of Bangladesh on the world map, highlighted in the red box, b) *boro* rice area published by Tiwari et al., [48], and the sample points from crop cut yield collected for the year 2019–2020 (blue circle) and 2020–2021 (maroon circle) respectively. The bold black lines represent crop zones, and the narrow gray lines represent district boundaries. The shapefile reprinted from GADM database under a CC BY license, with permission from Global Administrative Areas (www.gadm.org), original copyright 2018.

## 2.2 Dataset used

**2.2.1. Crop calendar.**   The Crop Calendar is a tool that provides information on the transplantation (sowing), growing, and harvest times of crops. The growing stage is particularly important for estimating *boro* rice yields. The *boro* rice crop calendar was developed by the Bangladesh Agriculture Research Council (BARC) and hosted on the Bangladesh Agro-Meteorological Information System (BAMIS) (Fig 2A and 2B). This calendar divides Bangladesh into fourteen distinct zones called "crop zones." This crop calendar was employed to identify the dates of the growing stage for *boro* rice. These dates were then used to select satellite datasets that covered the extent of the *boro* rice crop growing season.

**2.2.2. Reference data.**   District-level rice yield data for *boro* rice was obtained as published by the Bangladesh Bureau of Statistics (http://www.bbs.gov.bd/site/page/453af260-6aea-4331-b4a5-7b66fe63ba61/Agriculture), a government agency in Bangladesh. The data represented rice yields for each district and were measured in units of metric tons per hectare (tonn/ha). The district level yield data from 2006 to 2021 from all districts were downloaded and subsequently employed to train and validate the RF machine learning model. Additionally, crop yield data for rice were obtained through a collaborative effort involving the International Maize and Wheat Improvement Center (CIMMYT), the Bangladesh Agricultural Research Council (BARC), and the Bangladesh Rice Research Institute (BRRI) for *boro* rice crops. The crop-cut data measure the yield of specific rice crops in particular fields. This is accomplished by physically harvesting and weighing the crops to assess their productivity. The crop yield data was collected during the years 2019–2020 (n = 2,946) and 2020–2021 (n = 1845), specifically focusing on five districts (Dinajpur, Rajshahi, Khulna, Jashore and Rangpur) within Bangladesh (Fig 1B). These districts were chosen due to the availability of qualified scientific staff in these districts that could facilitate an ambitious yield measurement program from farmer's fields. Unfortunately, other districts were out of scope for grant funds to permit additional crop yield estimates. This dataset was a validation resource for the rice yield maps developed at 1,000 meters spatial resolution. However, this dataset was not incorporated into the training of

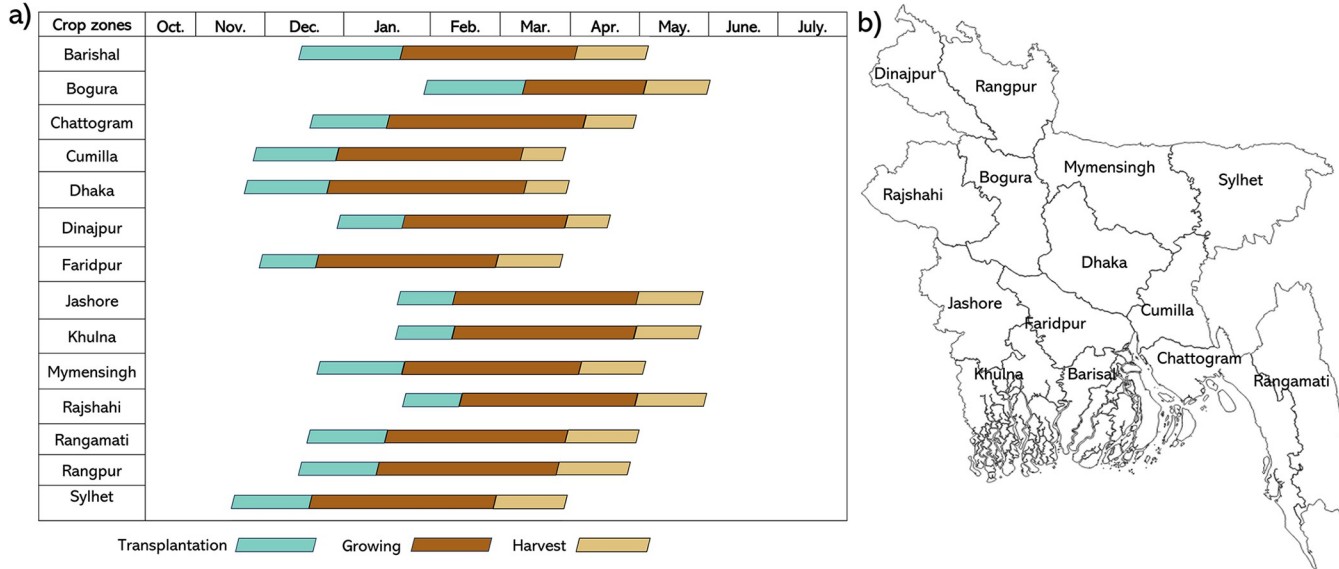

**Fig 2.**  a) The crop calendar of *boro* rice across all fourteen crop zones of Bangladesh (sourced from https://www.bamis.gov.bd/calendar), b) the unique crop zones of Bangladesh (https://www.bamis.gov.bd/calendar). The shapefile reprinted from GADM database under a CC BY license, with permission from Global Administrative Areas (www.gadm.org), original copyright 2018.

the RF model because the data needed to encompass all districts and, therefore, lacked variability. Moreover, the field collected data struggled to adequately capture the inherent variability present in rice yield patterns, a factor that could potentially limit the performance of RF regression models [49].

**2.2.3. Satellite data.** Remote sensing data were obtained from the MODIS, MOD09GA.061 product, Terra Surface Reflectance Daily Global data (surface reflectance band 1–7) at a spatial resolution of 500 m available on Google Earth Engine (GEE) [50]. The surface reflectance quality assurance band (QC_500m) was employed to identify and exclude bad pixels (cloudy, hazy pixels) from the surface reflectance dataset. Daily data was used from January to April from 2002 to 2021. The January to April time frame was specifically selected as this was the main *boro* rice growing season in all crop zones in Bangladesh (Fig 2A) [5]. Considering the emphasis on characterizing the overall surface state during broadly defined growing periods rather than relying on traditional phenological analysis, the daily data were transformed into 16-day mean composites for each year, resulting in 8 composite images per year. This was done because crop phenology does not change daily, and previous studies have used 16-day composite data for rice monitoring-related studies. Furthermore, using 16-day composites aligns with the methodology employed in previous rice monitoring-related studies [51, 52].

Vegetation and other indices were calculated, such as NDVI, EVI, Simple Ratio Water Index (SRWI), Land Surface Water Index (LSWI), Simple Ratio Tillage Index (SRTI), Normalized Difference Tillage Index (NDTI), Crop Residue Cover Index (CRCI), Modified CRC index (MCRC), SAVI, and Normalized Difference Senescent Vegetation Index (NDSVI) (Table 1), which are relevant and have shown importance in yield estimation in the past [53, 54]. In addition, the LST_Day_1km band from the MODIS MOD11A1 product was used for the same period (January to April), from 2002 to 2021. This data provides daily Land Surface Temperature (LST) at 1,000-meter spatial resolution. Bad pixels were masked using the QC_Day band to ensure data accuracy, and the LST values were subsequently converted from Kelvin to degrees Celsius. The mean monthly LST was computed for the period (January to April 2002 to 2021). The MODIS surface reflectance bands and indices were resampled to a 1000-meter spatial resolution and stacked together with MODIS LST to derive seven surface reflectance bands, ten indices, and one LST band, equivalent to eighteen bands per composite. These images were stacked together to contain seven surface reflectance bands, ten indices, and one LST band, equivalent to eighteen bands per composite. European Space Agency

**Table 1. Indices calculated using MODIS bands (B1-B7) which were used to estimate *boro* rice yields, where MODIS bands relate to indices as follows: B1 = RED, B2 = NIR1, B3 = BLUE, B4 = GREEN, B5 = SWIR1, B6 = SWIR2.**

| Indices | Full Name | Formula | References |
|---------|-----------|---------|------------|
| NDVI | Normalized Vegetation Index | NDVI = (NIR—RED) / (NIR + RED) | Gorten et al., [56] |
| EVI | Enhanced Vegetation Index | EVI = 2.5 * ((NIR1—RED)/ (NIR1 + 6 * RED—7.5 * BLUE +1) | Bolton at al., [57] |
| SRWI | Simple Ratio Water Index | NIR1/NIR2 | Basso et al., [58] |
| LSWI | Land Surface Water Index | (NIR1—SWIR1) / (NIR 1+SWIR1) | Chandrasekar et al., [59] |
| SRTI | Simple Ratio Tillage Index | SWIR1/SWIR2 | Hatfield et al., [60] |
| NDTI | Normalized Difference Tillage Index | (SWIR1-SWIR2)/ (SWIR1-SWIR2) | Memon et al., [61] |
| CRCI | Crop Residue Cover Index | (SWIR1 -BLUE)/ (SWIR1 + BLUE) | Gao et al., [62] |
| MCRC | Modified CRC index | (SWIR1 -GREEN)/ (SWIR1 + GREEN) | Bannari et al, [63] |
| SAVI | Soil Adjusted Vegetation Index | SAVI = (NIR1-RED) *(1 + 0.5)/(RED+NIR1 + 0.5) | Venancio et al., [64] |
| NDSVI | Normalized Difference Senescent Vegetation Index | (SWIR1—RED)/(SWIR1 + RED) | Hill et al., [65] |

(ESA) World Cover 2020, with an overall accuracy of 75% and a spatial resolution of 10 m [55], was used to generate a cropland mask. The cropland mask was resampled to 1,000 meters to match the MODIS resolution. Subsequently, it was used to exclude non-agricultural pixels from MODIS composite data and indices, which could introduce noise into the RF models.

## 2.3 Methodology

The methodology comprises three key steps: the establishment of the RF model, an evaluation of accuracy, and an analysis of yield trends.

**2.3.1 Setting up, random forest machine learning model.** An RF machine learning model was employed to estimate the *boro* rice yields (Fig 3). RF is a supervised ensemble method that leverages a collection of decision trees and utilizes the bagging technique (bootstrap and aggregation). Instead of relying solely on individual decision trees, the RF approach aggregates the results of multiple trees to determine the final output [65].

We chose the RF model because it has demonstrated its capabilities in crop yield estimation, including rice yield estimation [66–68]. This is due to the RF model's ability to capture the nonlinear relationships between biophysical parameters (input data) and crop yields more effectively than other ML-based models. Additionally, RF is robust to noise, which is crucial since satellite data and crop yield data often contain noisy information that can lead to inaccurate estimates. Furthermore, RF helps prevent overfitting. These factors make the RF model an ideal choice for crop yield estimation. Zonal statistics were computed to determine the district's mean values, considering both spatial and temporal dimensions, for all the bands in the stacked composite (MODIS bands and indices). This was done within 16-day composites and monthly LST measurements from January to April, covering 2006 to 2021 (with input variables = 148) across all 64 districts.

The variables in the stacked composites were used as input variables 'x' for the RF model. Simultaneously, the corresponding district-level *boro* rice yield data from 2006 to 2021, as published by BBS, were employed as the output variable ('y') in the RF model. In total, 960 observations were used in the RF model for training and testing.

Initially, the RF model was fitted with all 148 input variables ('x') and 960 observations in the stacked composites to perform hyperparameter tuning, employing the grid search method [66]. Hyperparameter tuning was used to enhance the effectiveness of an RF regression model. This process encompasses systematically adjusting critical model parameters to improve its predictive capabilities while minimizing the risk of overfitting. The parameters subjected to this optimization included variables such as the number of trees, tree depth, and the rate at which features are subsampled. Upon identifying the optimal parameters, the RF model was rerun, and model training and testing were carried out using the k-fold cross-validation technique to estimate district-level rice yields. However, we also test additional cross-validation techniques like hold-one-year-out, leave-one-out, and 70:30 train-test splits (S2 Table). A value of k = 5 was used in the k-fold validation technique, where k denoted the number of subsets into which the dataset was divided [67]. During each iteration, distinct subsets were designated as the test set, while the remaining data served for model training. After evaluating the performance of the RF model using accuracy metrics (Section 2.3.2), we calculated the Gini index to evaluate the effectiveness of splits and determine the relative importance of predictor variables in explaining the variability of the target variable. The top 30 input variables with the highest rankings in the Gini index were selected to re-run the RF forest regression model. The selection process for these 30 input variables involved iterative steps of the RF model, incrementing by five each time, starting from the top 10 variables. The model accuracy remained consistent with the initial accuracy (% RMSE) achieved using all 148 input variables. Finally,

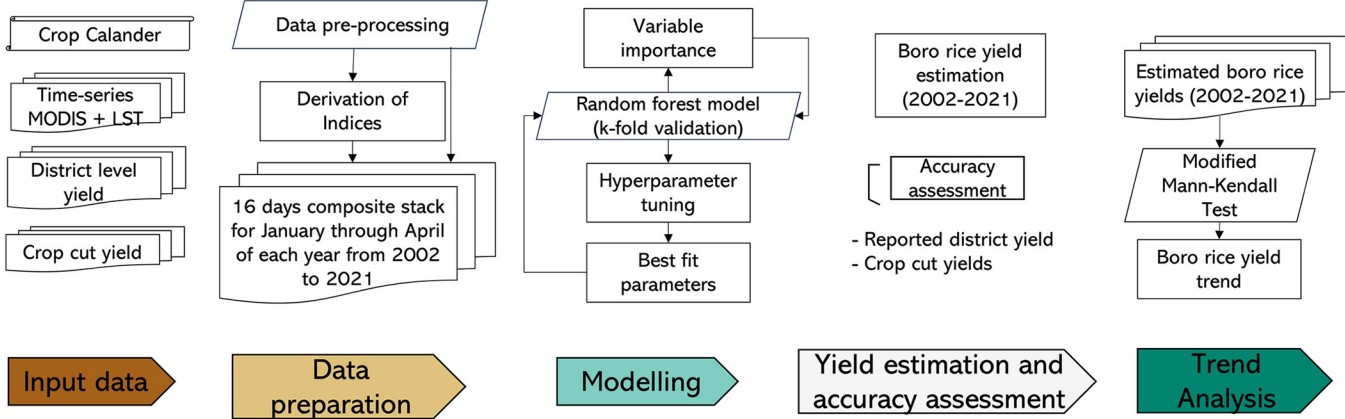

**Fig 3. A flow chart of the methodology applied to estimate *boro* rice yield in Bangladesh.** The process includes stages for gathering input data (brown), preparing the data (mustard), modeling the yield (sea green), estimating yield and assessing accuracy (gray), and conducting trend analysis (dark green).

the trained RF model and the top 30 input variables were used to estimate *boro* rice yield maps from 2002 to 2021.

**2.3.2 Accuracy assessment.**    Standard accuracy metrics, widely utilized in remote sensing studies, were used to assess the accuracy of the RF regression model [21, 68]. These included root mean squared error (RMSE), percentage root mean squared error (% RMSE), mean square error (MSE), and coefficient of determination ($R^2$). These statistical metrics were employed in the 5-fold cross-validation technique, and the mean values of these metrics were calculated to evaluate the accuracy of the RF model in 5-fold cross-validation. In addition, we calculated the uncertainty associated with the estimated *boro* rice yield map by estimating the uncertainty of predictions made by an RF ensemble. This estimation appears to be based on the standard deviation of predictions from individual trees in the ensemble [69]. Furthermore, the accuracy of the sub-district scale yield map (1,000 m spatial resolution) was assessed for the years 2019–2020 and 2020–2021 using crop cut yield data and the same statistical metrics (Fig 1B).

**2.3.3 Trend analysis.**    A trend analysis was conducted on the yearly RF-modeled *boro* rice yield from 2002 to 2021, employing the modified Mann-Kendall (MK) trend analysis method. Modified MK is a non-parametric trend test used to detect trends or monotonic patterns in time series data by adjusting for autocorrelation present in the data. The modified MK test was chosen over the MK trend test due to its susceptibility to positive annual autocorrelation in data [70]. This susceptibility heightens the risk of detecting trends that might not truly exist, thus favoring the modified version for enhanced accuracy [71]. Additionally, we calculated significant upward and downward trends at a 95% confidence interval (p-value < 0.05). This analysis enabled us to determine the areas with significant increasing and decreasing trends in *boro* rice yield.

## 3. Results

### 3.1 Hyperparameter tuning results and Gini index

The importance of all input variables in the RF model, computed using the Gini index, is shown in S1 Table. The 30 input variables that exhibited the highest Gini index values in the RF model are given in (Fig 4). Indices calculated from the February composites (SRWI, LSWI, Surface reflectance band 6) appeared high in the ranking, indicating their strong influence on the RF regression estimations. In contrast, data from April showed a lower Gini index, suggesting that they have a lesser impact on the target variable (*boro* rice yield) than February data.

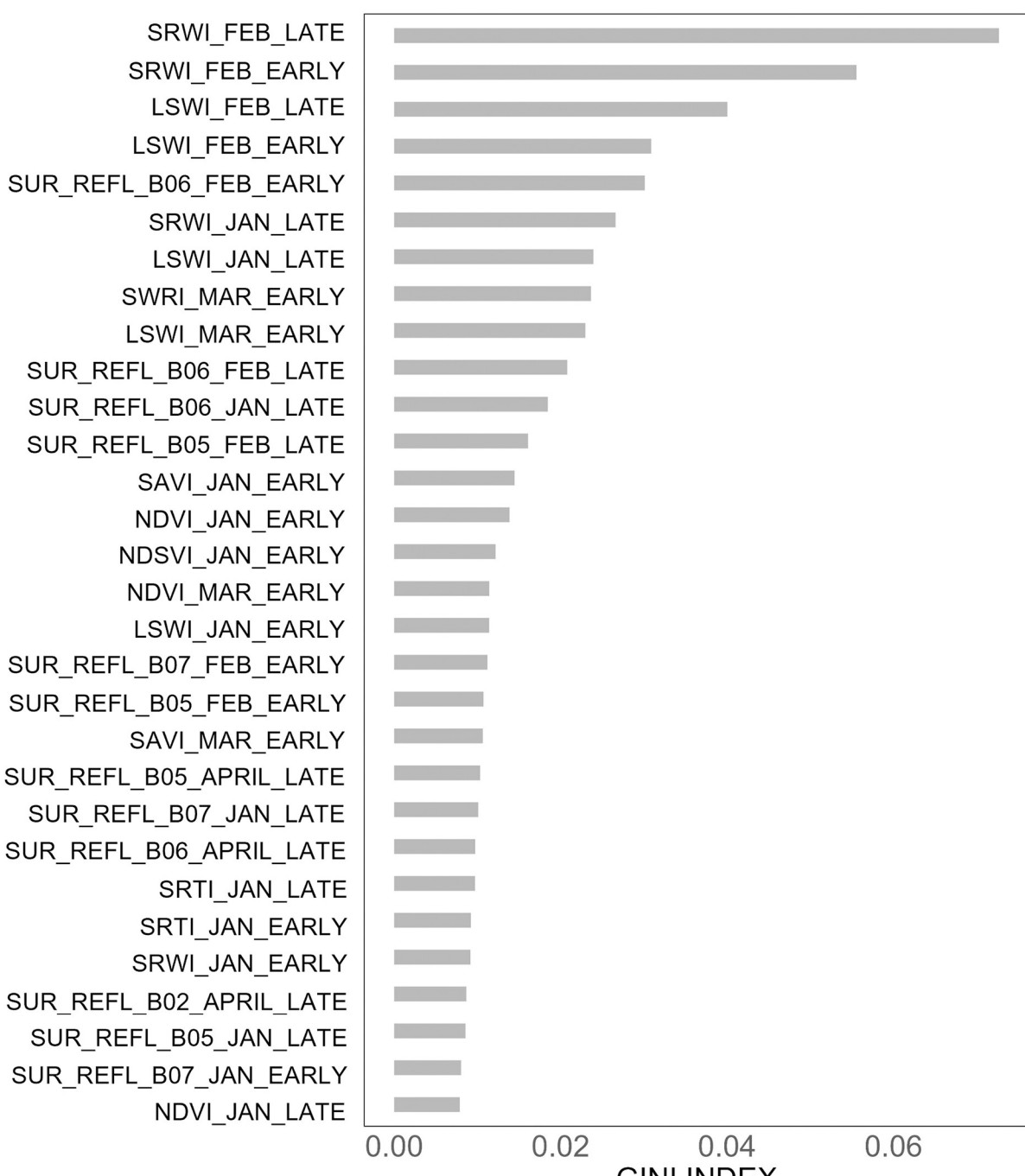

**Fig 4. Random forest top 30 variable importance utilizing all input variables, including the MODIS bands, spectral indices, and land surface temperature.** In the naming convention for MODIS bands and indices, the term 'Early' pertained to the 15-day composite image representing the first half of the month (days 1–16), while 'Late' pertained to the second half of the month (days 17–31).

### 3.2 Accuracy assessment

**3.2.1 District-level *boro* rice yield.**   We used a range of accuracy metrics when employing various input variables in the RF regression model with k-fold validation based on the Gini index ranking. These metrics encompassed an R2 value ranging from 0.43 to 0.57, MSE ranging from 0.12 to 0.09 ton/hectare, RMSE ranging from 0.31 to 0.35 ton/hectare, and percentage

**Table 2. Accuracy of the random forest regression model with various input variables (all inputs vs. top 30 inputs).**

| Model Configuration | Mean Square Error (MSE) (ton/hectare) | RMSE (ton/hectare) | % RMSE | R- Square ($R^2$) |
|---|---|---|---|---|
| RF (top 10 inputs) | 0.12 ± 0.02 | 0.35 ± 0.03 | 9.23 | 0.43 ± 0.06 |
| RF (top 15 inputs) | 0.10 ± 0.02 | 0.32 ± 0.03 | 8.4 | 0.53 ± 0.07 |
| RF (top 20 inputs) | 0.10 ± 0.02 | 0.31 ± 0.03 | 8.26 | 0.54 ± 0.10 |
| RF (top 25 inputs) | 0.10 ± 0.02 | 0.31 ± 0.03 | 8.22 | 0.54 ± 0.07 |
| RF (top 30 inputs) | 0.10 ± 0.02 | 0.31 ± 0.03 | 8.14 | 0.55 ± 0.07 |
| RF (all inputs) | 0.09 ± 0.02 | 0.31 ± 0.03 | 8.08 | 0.57 ± 0.07 |

RMSE spanning from 8.08% to 9.09% (Table 2). In the scenario where all input variables were taken into account (RF all inputs), the RF model demonstrated the lowest Mean Squared Error (MSE) (0.09 ± 0.02 ton/hectare), Root Mean Squared Error (RMSE) (0.31 ± 0.03 ton/hectare), and Percentage RMSE (% RMSE) (8.08). Conversely, when focusing solely on the top 10 inputs, the RF model exhibited high MSE 0.12 ± 0.02 ton/hectare), RMSE (0.34 ± 0.03 ton/hectare), and % RMSE (9.03%). Regarding the coefficient of determination ($R^2$) values, the RF model achieved its highest $R^2$ (0.57 ± 0.07) when utilizing all input variables, while the lowest $R^2$ values (0.45 ± 0.06) were observed when considering only the top 10 inputs. Additionally, a consistent trend was observed; as the number of input variables increased, particularly those with higher Gini index rankings, there was a reduction in % RMSE (decreasing from 9.03% to 8.08%), RMSE (decreasing from 0.35 to 0.31 ton/hectare), and MSE (going down from 0.12 to 0.09 ton/hectare), accompanied by an increase in $R^2$ values (rising from 0.45 to 0.57).

For the RF model utilizing the top 30 inputs, the values of MSE (0.10 ton/hectare), RMSE (0.31 ton/hectare), % RMSE (8.17%), and $R^2$ (0.55) exhibited minimal change compared to the RF model incorporating all inputs, which had values of MSE (0.09 ton/hectare), RMSE (0.31 ton/hectare), % RMSE (8.08%), and $R^2$ (0.57). This suggested that not all input variables were necessary for accurately estimating rice yields. Furthermore, the analysis revealed that images taken in February, January, and March held significant importance within the RF regression model (Fig 4). Utilizing images from these months facilitated midseason estimation of *boro* rice yield. Additionally, we found that the accuracy of the Random Forest regression method is not affected by employing different cross-validation techniques (hold-one-year-out, leave-one-out, and 70:30 train-test splits, S2 Table).

**3.2.2 Crop cut *boro* rice yield.** When employing the crop cut data, accuracy metrics were computed using RF (top 30 inputs) separately for five districts in Bangladesh. The MSE ranged from 0.52 to 0.85 ton/hectare, while RMSE spanned from 0.72 to 0.92 ton/hectare, and the percentage RMSE ranged between 11.12% and 14.22% for 2019–2020. Conversely, during 2020–2021, the MSE ranged from 0.54 to 0.98 ton/hectare, while RMSE spanned from 0.73 to 0.99 ton/hectare, and the percentage RMSE ranged between 11.45% and 15%. Furthermore, the lowest percentage of RMSE was observed in Faridpur (11.45%) for 2020–2021 and Khulna (11.12%) for 2019–2020. In contrast, the highest percentage of RMSE was noted in Rangpur (15%) in 2020–2021 and Dinajpur (14.22%) in 2019–2020. While the majority of districts exhibited comparable percentage RMSE values, the variance in percentage RMSE between the years 2019–2020 and 2020–2021 was least for Rangpur (2.28%) and Rajshahi (2.61%) (Fig 5).

## 3.3 Rice yield maps

Rice yield maps were produced from 2002 to 2021 (Fig 6). This was accomplished using a trained RF model that utilized the first 30 input variables with the highest variable importance (Gini index, Fig 4). Most of the areas in Bangladesh showed *boro* rice yield between 2–4 tons/hectare across years. Relatively low *boro* rice yield (0–2 ton/hectare) areas were observed in the northeast and southern regions of the country. Some areas randomly distributed across

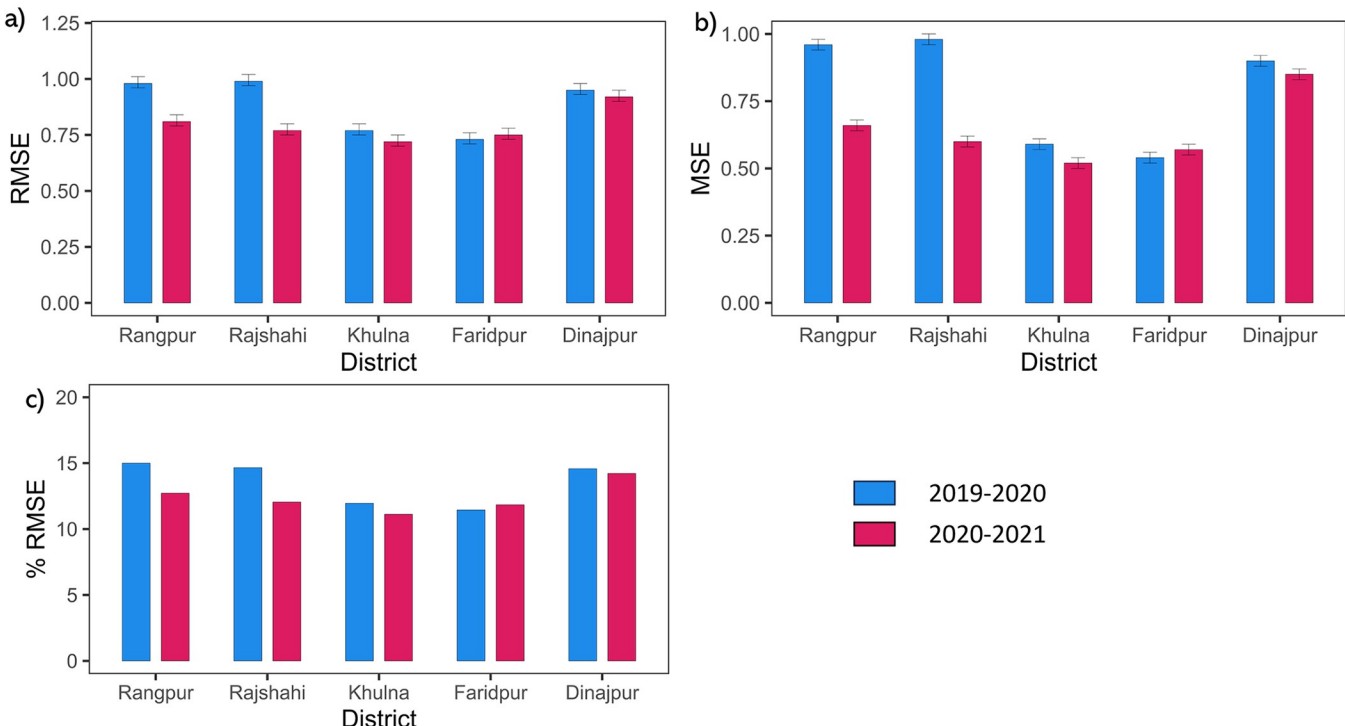

**Fig 5.** District-wise accuracy matrices of the random forest model computed using crop cut points collected for 2020 and 2021, including a) the Root Mean Squared Error (RMSE), b) the Mean Square Error (MSE), and c) the percentage Root Mean Squared Error (%RMSE).

Bangladesh showed high yield values (4–6 tons/hectares). In addition, *boro* rice yields observed in later years (2018–2021) were greater than yields from the early years (2002–2005). A persistent pattern of low yield values was noted in Bangladesh's northeastern and southern regions throughout the timeline. Mean (2002–2021) uncertainty of the estimated *boro* rice yield was also calculated. The average uncertainty was within the range of 0.40 to 0.45 tons per hectare, and this uncertainty was distributed randomly across Bangladesh (Fig 8).

### 3.4 Trend analysis maps

**3.4.1 *Boro* rice yield trend.**   A trend map was derived for rice yields from 2002 to 2021 using the modified MK trend test (Fig 7). The *boro* rice yield map was derived using a 95% confidence interval (CI) with a significance level of $p < 0.05$. In Bangladesh, there has been an increase in *boro* rice yields across most regions, contributing to 23.36% of the rice-planted area. However, several specific districts (Natore, Kishoreganj, Netrokona, Khulna, Feni, Dhaka, Narayanganj, Sirajganj, Sylhet, Sunamganj, Barguna, Patuakhali) have exhibited hotspots with declining trends in *boro* rice yields, contributing to 0.11% of the rice-planted area. Conversely, 76.51% of the randomly distributed rice-planted area in Bangladesh shows no trend in *boro* rice yields. The declining yield trend was not captured when the modified MK trend test was applied to district-level yields (S1 Fig).

## 4. Discussion

### 4.1 Performance of RF models for *boro* rice yield estimation

The RF regression exhibited high accuracy (%RMSE = 8.08%, $R^2$ = 0.57) when predicting rice yields by utilizing all input variables using reported *boro* rice yields. Interestingly, even when

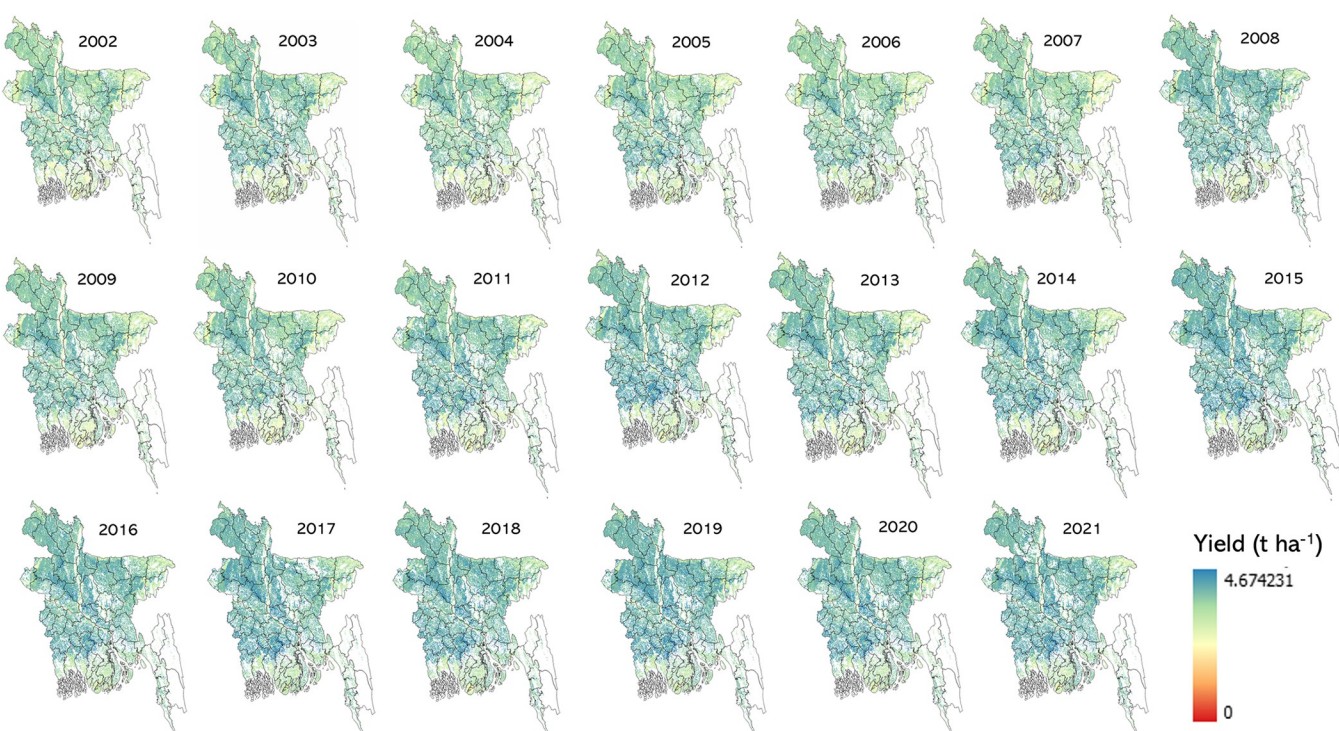

**Fig 6.** ***Boro* rice yield (2002–2021) estimation utilizing MODIS data and a trained random forest model with 30 input variables having high importance.** The shapefile reprinted from GADM database under a CC BY license, with permission from Global Administrative Areas (www.gadm.org), original copyright 2018.

reducing the input variables to 30, the RF regression achieved a %RMSE of 8.17% and $R^2$ = 0.55 when validation was done using district-level yearly data from 2006–2021 (Table 2). These 30 input variables included data from images collected in January, February, and March, indicating the feasibility of mid-season *boro* rice yield estimation (Fig 3). This approach provides a sufficient lead time of 1–2 months before *boro* rice yield can be reported, which could significantly contribute to the decision-making process related to food security. Uncertainty in the *boro* rice yield estimation varied between 0.40–0.45 tons/ hectares (Fig 8) and could be attributed to multiple factors. Firstly, it may result from errors in the district-level yield data reported by the Bangladesh Bureau of Statistics (BBS) from 2007 to 2021. These yields were utilized in the RF model and obtained through farmer interviews or crop cutting. Secondly, using a one-year agriculture mask, derived from the 2020 ESA land cover map, assumed that all crops planted in agricultural regions were rice. However, other crops were also present, albeit covering less than 10% of the area, and they can influence surface reflectance values, potentially introducing noise into the RF model. Thirdly, interannual variations from 2006 to 2021 in the agriculture mask could also introduce uncertainty into the RF model results. A rice crop calendar was used to divide Bangladesh into fourteen crop zones, each with unique rice sowing, peak, and harvest times. However, it is important to note that there were likely variations in the crop calendar at the district level, leading to differences in the timing of sowing, peak growth, and harvest for rice crops. Therefore, these district-level variations in the rice crop calendar must also be considered when accounting for uncertainty in rice yield estimations.

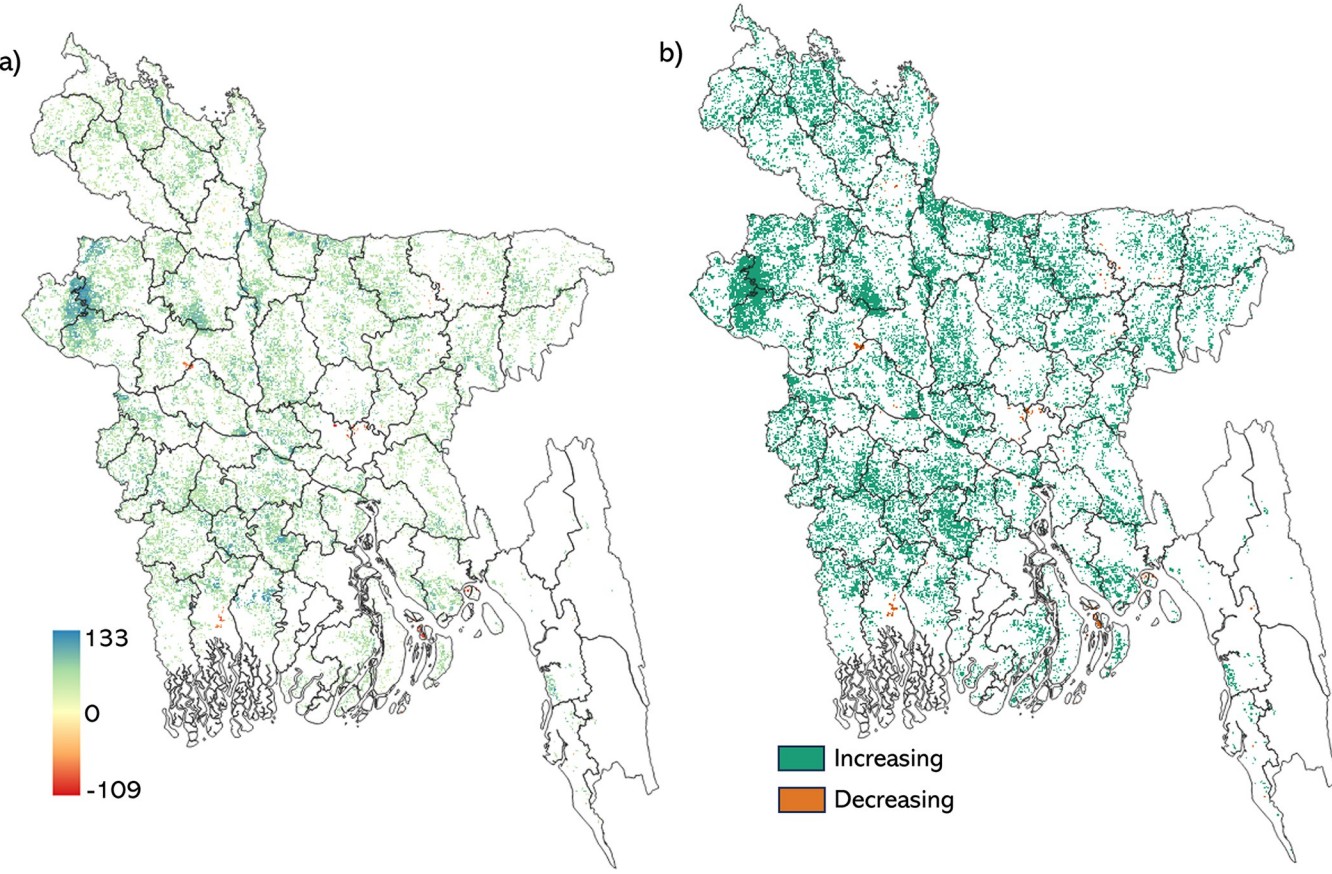

**Fig 7.** Results of a rice yield trend analysis across Bangladesh, including a) the magnitude of the modified MK score for rice yield trend from 2002–2021 with 95% CI (p value<0.05) and b) the classified map showing areas with increasing (green), and decreasing (orange) *boro* rice yield trends (2020–2021). The shapefile reprinted from GADM database under a CC BY license, with permission from Global Administrative Areas (www.gadm.org), original copyright 2018.

### 4.2 Validation of *boro* rice yields using crop cut data

When the estimated rice yields were validated using district-level crop cut data collected for the years 2019–2020 and 2020–2021, the % RMSE increased from 8.17% to 12.96%. This increase in % RMSE by 4.79% could be attributed to several factors. Firstly, these points might not have been accurately sampled from the correct locations, possibly due to uncertainties during the crop-cut experiments. Additionally, it's important to note that MODIS data and other biophysical parameters have a coarser spatial resolution, leading to mixed pixels within the rice fields [72]. Despite the coarser spatial resolution, the effects of mixed pixels would contribute less compared to uncertainties in collecting crop-cut data, as rice fields are homogeneously distributed, with 90% of the plantation being rice in Bangladesh.

Rajshahi and Rangpur exhibited high variability in % RMSE (variation of % RMSE = 2.44%) for the years 2019–2020 and 2020–2021. This variability is primarily due to the different number of samples used during these two years. Variations in sample size, as well as differences in sampling design, can significantly impact the accuracy of satellite-based crop yield estimation [73]. For Rajshahi, 884 crop cut samples were used for 2019–2020 and 58 for 2020–2021, and for Rangpur, 839 samples for 2019–2020 and 172 samples for 2020–2021 were used. On the other hand, the districts that have nearly the same number of crop cut samples (average percentage difference in sample points = 25.79%) show less variation (average

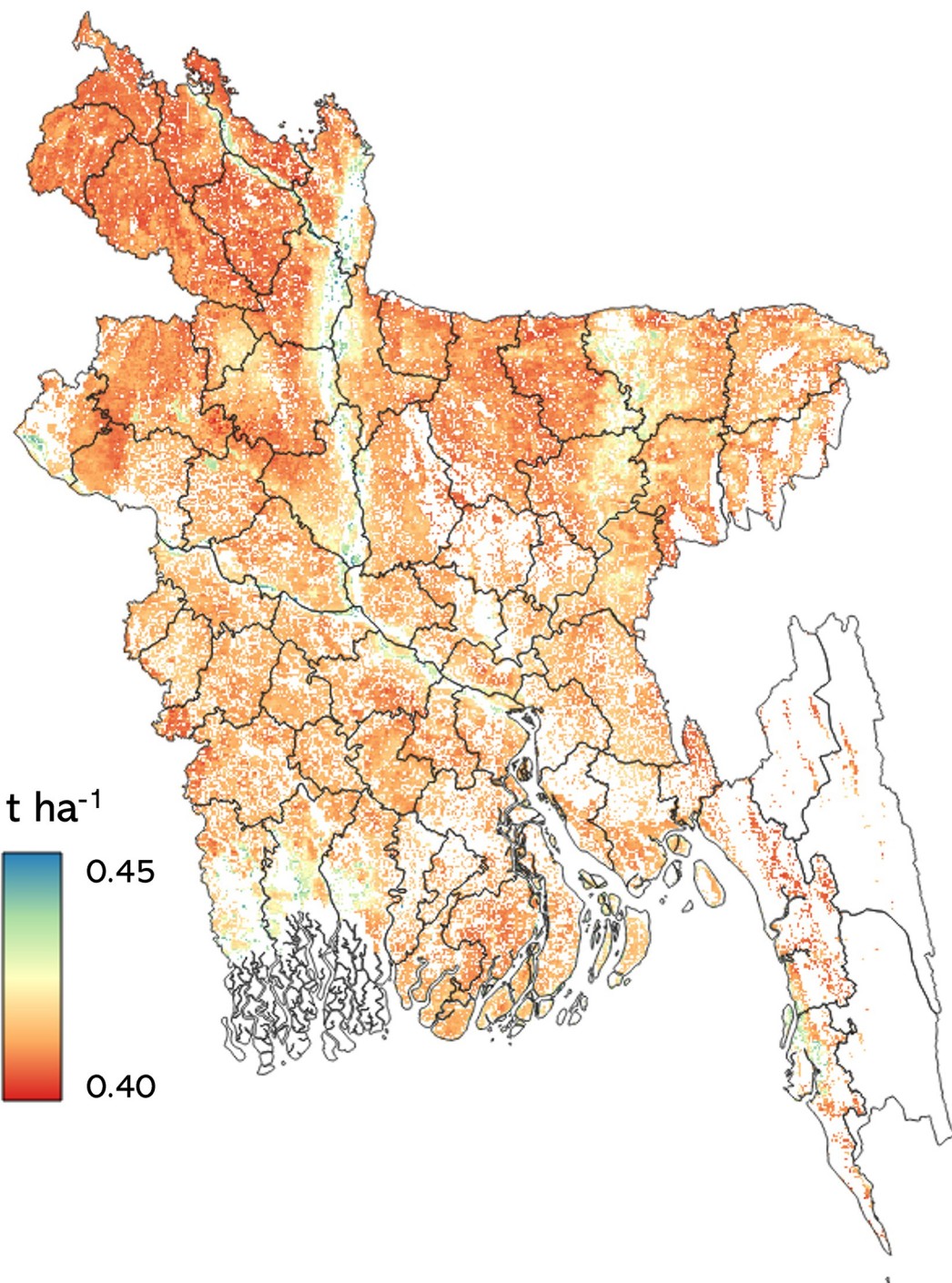

**Fig 8. Mean uncertainty of *boro* rice yields estimated by random forest model from 2002–2021.** The shapefile reprinted from GADM database under a CC BY license, with permission from Global Administrative Areas (www.gadm.org), original copyright 2018.

variation % RMSE = 0.27%) in % RMSE. Khulna is an exception where there was an exceptionally high difference in the number of points between the two years (average percentage difference in sample points = 306%), and the percentage RMSE difference was 0.84% (Fig 5).

Improvements to the field sampling strategy could lead to better satellite-based crop yield estimation [68, 73, 74]. In this study, *boro* rice crop yield data was collected from five districts in Bangladesh—Dinajpur, Rajshahi, Khulna, Jashore, and Rangpur (Fig 1B)—due to the availability of data in these regions. Data from other districts were beyond the scope of the grant funding and therefore could not be included. However, in the future, data from additional districts could be collected and used for validation in a similar manner.

## 4.3 Comparison of rice yields with previous studies

The accuracy of rice yield estimation was compared with data from previous studies. For example, Alam et al. 2019 utilized various regression methods (lasso, decision tree, ordinary linear regression) to estimate yield in specific regions of Bangladesh, achieving a high mean $R^2$ of 0.64 between reported and estimated *boro* rice yield in contrast to our study where the R2 was 0.56. However, it's worth noting that they did not report RMSE, an essential statistical parameter for evaluating the uncertainty in rice yield estimation. Furthermore, their estimation of *boro* rice yield was conducted at a coarser spatial scale, specifically at the division level, in contrast to the 1,000 m rice yield maps developed herein. Similarly, Mosleh et al., used MODIS data and a regression method to estimate yield for 2010–2012, employing linear regression, and found the RMSE and $R^2$ to be 0.25 Mton/ha and 0.81, respectively [38]. However, they estimated *boro* rice yield at the district level and for only two years (2010–2012), so the approach's applicability may be questionable for yield estimates over longer periods. In the study by Islam et al., Sentinel 2 data were employed at the sub-district level to predict yields in the northeastern region of Bangladesh using an Artificial Neural Network (ANN) approach [42]. The study reported $R^2$ values of 0.83 between measured and modeled yields. It is worth noting that their model contained region-specific characteristics, suggesting potential limitations when applied to other regions within Bangladesh. Furthermore, the analysis was based on data spanning only three years (2017–2019), which could introduce uncertainties when estimating yields over longer periods. While other studies have achieved better model performance statistics, the present study is the first to map rice yield across Bangladesh at a 1,000 m spatial resolution over a period spanning two decades. This mapping was further validated using reported district-level yields and crop-cut sample points collected over two years in five districts. The uncertainty of the *boro* rice estimates was also quantified in this study, a gap that was not addressed in the previous studies.

The estimated *boro* rice yield trends were also compared with previous studies [43, 75–77]. *Boro* rice yield showed an increasing trend from 2002–2021 for most of the areas, which others demonstrated [78–80]. The increasing trend in *boro* rice production could be attributed to hybrid varieties with shorter crop cycles compared to local rice. This could result in a higher *boro* rice yield (6.0–7.5 t/ha), provided sufficient water available for irrigation and optimal air temperatures during the *boro* growing season [43]. In this study, negative trends in *boro* rice yields were observed in some areas, potentially attributed to the adverse effects of temperature, as indicated by previous research [12, 81]. Whereas the declining rice yield trend in the coastal region could be attributed to saltwater intrusion, which harms rice yields, as brought up by Islam et al. [82]. The expected impact of climate change is even more concerning [83], with projected average *boro* yield reductions of over 20% and 50% by the years 2050 and 2070, respectively [76].

## 4.4 Research applicability

Various efforts have been made in the past to estimate *boro* rice yield in Bangladesh using remote sensing satellites and biophysical parameters [37–39, 84]. However, these studies were

typically limited to specific regions of Bangladesh, often with coarser spatial resolution and covering only a year at the district or sub-district level. Furthermore, these studies relied on regression models for yield estimation, which can't capture the non-linear relationship between independent and dependent variables.

In contrast, the present study utilized remote sensing data and machine learning, specifically an RF model, to estimate yields at a sub-district scale (1,000 m spatial resolution), spanning 2002 to 2021. The investigation has shown promising results with a low root mean square error (%RMSE = 8.07%). This study can be valuable for the Government of Bangladesh to estimate rice yields for future years. Additionally, the government can use this information to create detailed maps showing areas with increasing or decreasing *boro* rice trends and temperature and rainfall trends. This data can inform decisions to enhance *boro* rice production in the near future and adjust *boro* rice crop calendars to adapt to climate change. Furthermore, the *boro* rice yield map can help identify areas where drought-tolerant rice varieties could be introduced to boost *boro* rice production. Moreover, the Bangladesh Bureau of Statistics (BBS) could integrate this innovative rice mapping framework into their conventional *boro* rice yield estimation system to develop an automated national rice yield estimation and monitoring system, thereby contributing to the advancement of smart agriculture in Bangladesh.

## 5. Conclusion

In conclusion, spectral bands and vegetation indices derived from time-series satellite imagery (MODIS) were used along with an RF machine learning method to estimate *boro* rice yield at a sub-district scale (1,000 m spatial resolution). The methodology provided *boro* rice yield estimates during the peak season (before harvest) in Bangladesh's heterogeneous and diversified cropping systems. The spectral bands and indices, such as LSWI, SRWI, NDVI, and SAVI, as well as surface reflectance bands 2, 5, 6, and 7 from January, February, and March, exhibited higher rankings in the Gini index, indicating their importance in the RF model for *boro* rice yield estimation.

The approach, which relied on harnessing district-level reported yields and satellite data (MODIS), showed promising results (% RMSE = 8%) and proved to be effective in estimating *boro* rice yields at a sub-district scale (1,000 meters spatial resolution). Additionally, the *boro* rice yield trend observed from 2002 to 2021 in this study provided valuable insights into areas with decreasing and increasing yield trends. Furthermore, maps depicting rainfall and temperature trends for other studies could help identify regions where *borfo* rice yields are currently affected and pinpoint areas where rising temperatures and rainfall trends may negatively affect future *boro* rice yields. The methodology can provide independent, evidence-based information on *boro* rice yield estimates (winter rice), which is useful for stakeholders in Bangladesh for planning and implementing present and future policies related to food security management.

## 6. Future work

A 2021 agriculture mask, derived from an ESA land cover map, was utilized. This mask was particularly effective because 80–90% of the agricultural areas consisted of *boro* rice, which varies annually (Bangladesh Bureau of Statistics, 2020) [85]. However, this study used training data from different years (2006–2021) to estimate rice yields from different years (2002–2021). Therefore, using a single-year crop mask could introduce errors in estimating rice yield. In the future, unique rice masks should be used for each growing season [48, 86], which could reduce uncertainty in rice yield estimation. Future studies, however, could use Bangladesh's micro-

scale agroecological zone classifications (http://apps.barc.gov.bd) to improve the accuracy of rice mapping, provided rice yield data is available from sub-district scales or crop-cut methods. In this study, crop cut data from five districts were utilized for validation of estimated rice yields. However, following the approach discussed in this study, crop cut data from other districts could also be collected in the future to enhance the validation process.

High-resolution satellite data, such as Sentinel-2, could also generate high-resolution rice yield maps (10m). Implementing deep learning models with Long Short-Term Memory (LSTM) nodes, which excel at handling long-term dependencies and possess memory retention capabilities, could also improve rice yield estimates. However, substantial data are required to adequately train deep neural network models [87]. Our future work also aims to quantify the impact of climate change on *boro* rice production in Bangladesh. The *boro* rice yield maps generated for two decades could be instrumental in identifying regions where *boro* rice yields are affected by temperature and rainfall during different growing stages. Additionally, they can help pinpoint areas where *boro* rice yields might be negatively affected by rising temperature and rainfall trends in the near future.

## Supporting information

**S1 Table. Random forest variable importance utilizing all input variables, including the MODIS bands, spectral indices, and land surface temperature.** In the naming convention for MODIS bands and indices, the term 'Early' pertained to the 15-day composite image representing the first half of the month (days 1–16), while 'Late' pertained to the second half of the month (days 17–31).
(DOCX)

**S2 Table. Comparison between different validation methods in random forest classification.**
(DOCX)

**S1 Fig.** a) magnitude of the modified mk test on district level rice yields, b) district level trends in the rice yields.
(DOCX)

## Acknowledgments

The authors acknowledge the Department of Agricultural Extension (DAE) and CIMMYT staff who collected field data (sample points). The authors would also like to recognize the graduate school writing center at North Carolina State University for improving the quality of the paper. This research has also received backing from the United States Department of Agriculture (USDA). The work was conducted while the paper's primary author was participating in the USDA Data Science Internship Program.

This work was partially supported by the Bill and Melinda Gates Foundation and USAID through the Cereal Systems Initiative for South Asia (CSISA; https://csisa.org/) and the CGIAR Regional Integrated Initiative for Transforming Agrifood Systems in South Asia (TAFSSA;https://www.cgiar.org/initiative/20-transforming-agrifood-systems-in-south-asia-tafssa/). Accordingly, we thank all funders who supported this research by contributing to the CGIAR Trust Fund (https://www.cgiar.org/funders/). The views and opinions in this document are those of the authors and do not necessarily reflect those of the Gates Foundation, USAID, or CGIAR and shall not be used for advertising purposes.

## Author Contributions

**Conceptualization:** Varun Tiwari.

**Data curation:** Varun Tiwari, Mohammad Kamruzzaman, Timothy J. Krupnik.

**Formal analysis:** Varun Tiwari.

**Investigation:** Varun Tiwari, Kelly Thorp, Joshua Gray.

**Methodology:** Varun Tiwari, Kelly Thorp, Mirela G. Tulbure, Joshua Gray, Mohammad Kamruzzaman, Timothy J. Krupnik.

**Project administration:** Varun Tiwari.

**Software:** Varun Tiwari.

**Supervision:** Kelly Thorp, Mirela G. Tulbure, Joshua Gray, Timothy J. Krupnik, A. Sankarasubramanian.

**Validation:** Varun Tiwari.

**Visualization:** Varun Tiwari.

**Writing – original draft:** Varun Tiwari, Kelly Thorp.

**Writing – review & editing:** Mirela G. Tulbure, Joshua Gray, Mohammad Kamruzzaman, Timothy J. Krupnik, A. Sankarasubramanian, Marcelo Ardon.

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
