## [Decision Letter · Decision Letter 0]

11 Jul 2024

PONE-D-24-12288Mapping Rice Yield Using Remote Sensing and Machine LearningPLOS ONE

Dear Dr. Tiwari,

Thank you for submitting your manuscript to PLOS ONE. After careful consideration, we feel that it has merit but does not fully meet PLOS ONE’s publication criteria as it currently stands. Therefore, we invite you to submit a revised version of the manuscript that addresses the points raised during the review process.

We look forward to receiving your revised manuscript.

Kind regards,

Salim Heddam

Academic Editor

PLOS ONE

Journal Requirements:

3. Thank you for stating the following financial disclosure: "No. The funders had no role in study design, data collection and analysis, decision to publish, or preparation of the manuscript" 

4. We note that Figures 1, 2, 6, 7 and 8 in your submission contain [map/satellite] images which may be copyrighted. All PLOS content is published under the Creative Commons Attribution License (CC BY 4.0), which means that the manuscript, images, and Supporting Information files will be freely available online, and any third party is permitted to access, download, copy, distribute, and use these materials in any way, even commercially, with proper attribution. For these reasons, we cannot publish previously copyrighted maps or satellite images created using proprietary data, such as Google software (Google Maps, Street View, and Earth). For more information, see our copyright guidelines: http://journals.plos.org/plosone/s/licenses-and-copyright.

a. You may seek permission from the original copyright holder of Figures 1, 2, 6, 7 and 8 to publish the content specifically under the CC BY 4.0 license.  

5. We notice that your supplementary tables are included in the manuscript file. Please remove them and upload them with the file type 'Supporting Information'. Please ensure that each Supporting Information file has a legend listed in the manuscript after the references list.

Additional Editor Comments:

Reviewer 1#:I have completed the review of the manuscript titled "Mapping Rice Yield Using Remote Sensing and machine Learning" submitted to PLOS ONE. I am pleased to recommend its acceptance for publication with minor revisions.

The manuscript presents a well-structured and clearly articulated study that addresses the crucial need for timely and accurate estimation of rice yields, particularly focusing on boro rice production in Bangladesh. The methodology employed in this research, leveraging MODIS data and machine learning techniques, demonstrates innovation and rigor in tackling the complexities of yield estimation at a subdistrict scale.

The authors have provided comprehensive details regarding the workflow developed for boro rice yield estimation, along with the validation process, which enhances the credibility of the findings. The inclusion of a trend analysis using the modified Mann-Kendall test further strengthens the robustness of the study's conclusions.

Minor revisions are suggested to improve clarity and readability. I have attached my comments in the below. Specifically, attention to minor typos and consistency in terminology usage throughout the manuscript would enhance its overall coherence. Additionally, ensuring the completeness of references and proper formatting according to the journal's guidelines is recommended.

Overall, the manuscript makes a significant contribution to the field of agricultural science and remote sensing applications in crop yield estimation. The findings have implications for food security management, agricultural policy development, and climate change adaptation not only in Bangladesh but also in other rice-producing regions globally.

I commend the authors for their thorough research and insightful analysis. With the implementation of the suggested minor revisions, I believe this manuscript will be well-suited for publication in PLOS ONE.

Thank you for the opportunity to review this manuscript.

General comments:

1. Check for the typos and blank space throughout the manuscript.

2. Please ensure all figures are high quality. I encourage the authors to submit high-resolution versions of the figures for publication.

3. The source of the shapefile used to create the figure is not mentioned in the text. Authors are encouraged to cite the source of the shapefile.

4. 2.2.2. Reference data: It's unclear whether these yield data are for rice or all crops. Additionally, the timeframe for which the yields were collected is not mentioned.

5. 2.2.3. Satellite data: Here, you mention using spectral bands at 500-meter spatial resolution and LST data at 1000-meter resolution. It's unclear whether you resampled the MODIS spectral data to 1000 meters to match the LST resolution.

6. 2.3.1 Setting up, random forest machine learning model: I'd recommend exploring additional cross-validation techniques like hold-one-year-out, leave-one-out, and 70:30 train-test splits. This can help assess the model's generalizability and consistency across different data partitions.

Good Luck!

Reviewer 2#:The paper estimate rice yield using remote sensing and machine learning model. The topic is interesting and covers an important issue of food security. However, this paper lacks novelty. I have some suggestions which may be addressed before its quality improvement. My comments are below:

1. Title is somewhat misleading. Random forest is a very commonly used in crop science.

2. In abstract, what is the gaps in earlier literature? what is the implication of this research.? Abstract might be comprehensively revised

3. Introduction section – weak rational justification. How this study can contribute in the literature? What is hypothesis? These are not clear. There are some related works used Random Forest model, for example Ghose et al. 2021.

4. In section 2.2, main weakness of the paper is data quality control. The crop yield data was collected during the years 2019-2020 (n= 2,946) and 2020-2021 (n=1845), specifically focusing on five districts (Dinajpur, Rajshahi, Khulna, Jashore and Rangpur) within Bangladesh. Why only five districts were considered for rice yield analysis for this paper?

5. K-fold valuation is used, but how many k-fold used for this study? Is there any resample technique used for this research?

6. Why authors have used only random forest technique and why not other machine learning techniques for this paper?

7. Discussion section is very poor. Scholarly discission is required. Authors should add limitation of the paper.

8. In conclusion, the authors may discuss how this study will be utilized for the policy and planning especially in the case of future food security.

Reviewers' comments:

Reviewer's Responses to Questions

**Comments to the Author**

1. Is the manuscript technically sound, and do the data support the conclusions?

Reviewer #1: Yes

Reviewer #2: No

2. Has the statistical analysis been performed appropriately and rigorously? 

Reviewer #1: Yes

Reviewer #2: No

3. Have the authors made all data underlying the findings in their manuscript fully available?

Reviewer #1: Yes

Reviewer #2: Yes

4. Is the manuscript presented in an intelligible fashion and written in standard English?

Reviewer #1: Yes

Reviewer #2: No

5. Review Comments to the Author

Reviewer #1: I have completed the review of the manuscript titled "Mapping Rice Yield Using Remote Sensing and machine Learning" submitted to PLOS ONE. I am pleased to recommend its acceptance for publication with minor revisions.

The manuscript presents a well-structured and clearly articulated study that addresses the crucial need for timely and accurate estimation of rice yields, particularly focusing on boro rice production in Bangladesh. The methodology employed in this research, leveraging MODIS data and machine learning techniques, demonstrates innovation and rigor in tackling the complexities of yield estimation at a subdistrict scale.

The authors have provided comprehensive details regarding the workflow developed for boro rice yield estimation, along with the validation process, which enhances the credibility of the findings. The inclusion of a trend analysis using the modified Mann-Kendall test further strengthens the robustness of the study's conclusions.

Minor revisions are suggested to improve clarity and readability. I have attached my comments in the below. Specifically, attention to minor typos and consistency in terminology usage throughout the manuscript would enhance its overall coherence. Additionally, ensuring the completeness of references and proper formatting according to the journal's guidelines is recommended.

Overall, the manuscript makes a significant contribution to the field of agricultural science and remote sensing applications in crop yield estimation. The findings have implications for food security management, agricultural policy development, and climate change adaptation not only in Bangladesh but also in other rice-producing regions globally.

I commend the authors for their thorough research and insightful analysis. With the implementation of the suggested minor revisions, I believe this manuscript will be well-suited for publication in PLOS ONE.

Thank you for the opportunity to review this manuscript.

General comments:

1. Check for the typos and blank space throughout the manuscript.

2. Please ensure all figures are high quality. I encourage the authors to submit high-resolution versions of the figures for publication.

3. The source of the shapefile used to create the figure is not mentioned in the text. Authors are encouraged to cite the source of the shapefile.

4. 2.2.2. Reference data: It's unclear whether these yield data are for rice or all crops. Additionally, the timeframe for which the yields were collected is not mentioned.

5. 2.2.3. Satellite data: Here, you mention using spectral bands at 500-meter spatial resolution and LST data at 1000-meter resolution. It's unclear whether you resampled the MODIS spectral data to 1000 meters to match the LST resolution.

6. 2.3.1 Setting up, random forest machine learning model: I'd recommend exploring additional cross-validation techniques like hold-one-year-out, leave-one-out, and 70:30 train-test splits. This can help assess the model's generalizability and consistency across different data partitions.

Good Luck!

Reviewer #2: The paper estimate rice yield using remote sensing and machine learning model. The topic is interesting and covers an important issue of food security. However, this paper lacks novelty. I have some suggestions which may be addressed before its quality improvement. My comments are below:

1. Title is somewhat misleading. Random forest is a very commonly used in crop science.

2. In abstract, what is the gaps in earlier literature? what is the implication of this research.? Abstract might be comprehensively revised

3. Introduction section – weak rational justification. How this study can contribute in the literature? What is hypothesis? These are not clear. There are some related works used Random Forest model, for example Ghose et al. 2021.

4. In section 2.2, main weakness of the paper is data quality control. The crop yield data was collected during the years 2019-2020 (n= 2,946) and 2020-2021 (n=1845), specifically focusing on five districts (Dinajpur, Rajshahi, Khulna, Jashore and Rangpur) within Bangladesh. Why only five districts were considered for rice yield analysis for this paper?

5. K-fold valuation is used, but how many k-fold used for this study? Is there any resample technique used for this research?

6. Why authors have used only random forest technique and why not other machine learning techniques for this paper?

7. Discussion section is very poor. Scholarly discission is required. Authors should add limitation of the paper.

8. In conclusion, the authors may discuss how this study will be utilized for the policy and planning especially in the case of future food security.

6. PLOS authors have the option to publish the peer review history of their article (what does this mean?). If published, this will include your full peer review and any attached files.

Reviewer #1: **Yes: **Md Abdur Rouf Sarkar

Reviewer #2: No

---

## [Author Response · Author response to Decision Letter 0]

11 Aug 2024

We are pleased to resubmit our revised manuscript titled " Advancing Food Security: A Framework for Rice Yield Estimation using Time Series of Optical Data and Machine Learning Algorithm (changed title based on reviewers suggestion)" for your consideration for publication in PLOS ONE. We have addressed all the comments related to the journal's requirements, as well as the feedback provided by the reviewers in their Word file. Additionally, we have prepared both a clean version and a tracked changes version of the revised manuscript, incorporating all of the reviewers' suggestions.

We have also ensured the following journal requirements are met:

We have made sure that the manuscript adheres to PLOS ONE's style guidelines, including those for file naming.

The code used in this work is available upon request.

The authors received no specific funding for this work. We would like to mention that the primary author received a salary from NC State University and utilized university resources for this work.

We have revised Figure 1 by removing the background Google Earth map. The maps included in Figures 2, 6, 7, and 8 were generated by the authors and, therefore, do not require any permissions or copyrights. However, the source of the data (shapefile used) is acknowledged in the figure captions. All PLOS content is published under the Creative Commons Attribution License (CC BY 4.0).

We have removed the supplementary material from the manuscript and included it as a separate file in the submission.

Thank you for considering our revised manuscript. We look forward to your feedback.

We would like to thank the reviewers for their valuable comments. Our responses are in Italics for each comment and the corresponding text is added in the track-changed and clean version of the manuscript. 

Reviewer 1

1. Check for typos and blank spaces throughout the manuscript.

Thanks for bringing this up. We have thoroughly checked the spelling, grammar, and blank spaces throughout the manuscript.

2. Please ensure all figures are high quality. I encourage the authors to submit high-resolution versions of the figures for publication.

All the figures are generated per the journal guidelines (300 dpi).

3. The source of the shapefile used to create the figure is not mentioned in the text. Authors are encouraged to cite the source of the shapefile.

Modified line Lines 114-116, 127-130, Line 296-298, Line 309-313, Line 334-336: The shapefile reprinted from the GADM database under a CC BY license, with permission from Global Administrative Areas (www.gadm.org), original copyright 2018.

Added this statement in Fig1,2,7 and 8: “Shapefile reprinted from GADM database under a CC BY license, with permission from Global Administrative Areas (www.gadm.org), original copyright 2018. The figure was made with QGIS 3.22.7 under a CC BY license.”

4. 2.2.2. Reference data: It's unclear whether these yield data are for rice or all crops. Additionally, the timeframe for which the yields were collected is not mentioned.

We clarify that this was only rice yield data and added in Lines 136-139: “Additionally, crop yield data for rice were obtained through a collaborative effort involving the International Maize and Wheat Improvement Center (CIMMYT), the Bangladesh Agricultural Research Council (BARC), and the Bangladesh Rice Research Institute (BRRI) for boro rice crops”

5. 2.2.3. Satellite data: Here, you mention using spectral bands at 500-meter spatial resolution and LST data at 1000-meter resolution. It's unclear whether you resampled the MODIS spectral data to 1000 meters to match the LST resolution.

Revised the sentence in Lines 169-171 “The MODIS surface reflectance bands and indices were resampled to a 1000-meter spatial resolution and stacked together with MODIS LST to derive seven surface reflectance bands, ten indices, and one LST band, equivalent to eighteen bands per composite.”

6. 2.3.1 Setting up, a random forest machine learning model: I'd recommend exploring additional cross-validation techniques like hold-one-year-out, leave-one-out, and 70:30 train-test splits. This can help assess the model's generalizability and consistency across different data partitions.

Thank you for your suggestion, we did the additional analysis and found that the accuracy of the Random Forest regression method is not affected by the cross-validation technique. Here is the summary of our results.

Line 207-208: “However, we also tested additional cross-validation techniques like hold-one-year-out, leave-one-out, and 70:30 train-test splits.”

Line 251-252: “We used a range of accuracy metrics when employing various input variables in the RF regression model with k-fold validation based on the Gini index ranking”

Line 269-271: “Additionally, we found that the accuracy of the Random Forest regression method is not affected by employing different cross-validation techniques (hold-one-year-out, leave-one-out, and 70:30 train-test splits, columns 3,4 and 5 Table S3).”

S3 Table. Comparison between different validation methods in random forest classification.

Accuracy metrics K-fold Ratio of the dataset (70:30) Leave-one-out Held-one-year-out

Average RMSE 0.31 0.28 0.22 0.31

Average R2 0.57 0.56 0.57 0.50

Average MSE 0.098 0.08 0.09 0.10

Average percentage RMSE 8.08 8.10 5.91 8.10

Reviewer 2

Reviewer 2#: The paper estimates rice yield using remote sensing and machine learning models. The topic is interesting and covers an important issue of food security. However, this paper lacks novelty. I have some suggestions which may be addressed before its quality improvement. My comments are below:

1. The title is somewhat misleading. Random forest is a very commonly used in crop science.

Thanks for bringing this up we have modified the title which now read as….Advancing Food Security: A Framework for Rice Yield Estimation using Time Series of Optical Data and Machine Learning.

2. In the abstract, what is the gaps in earlier literature? what is the implication of this research.? The abstract might be comprehensively revised

We have modified the abstract (changes in Italics), which now reads as:

Lines 15-36 "Timely and accurate rice yield estimation is crucial for supporting food security management, agricultural policy development, and climate change adaptation in rice-producing countries such as Bangladesh. However, a significant gap exists in the application of remote sensing methods for government-reported rice yield estimation for food security management at high spatial resolution. Current methods are limited to specific regions and primarily used for research, lacking integration into national reporting systems. Additionally, there is no consistent yearly boro rice yield map at a sub-district scale, hindering localized agricultural decision-making.

To address this need, this study introduced a workflow to enable timely and precise rice yield estimation at a sub-district scale (1,000-meter spatial resolution). This workflow leveraged MODIS and annual district-level yield data to train a random forest model for estimating boro rice yields from 2002 to 2021. The results revealed a mean percentage root mean square error (RMSE) of 8.07% and 12.96% when validation was conducted using reported district yields and crop-cut yield data, respectively. Additionally, the estimated yield of boro rice varies with an uncertainty range between 0.40 and 0.45 tons per hectare across Bangladesh.

Furthermore, a trend analysis was performed on the estimated boro rice yield data from 2002 to 2021 using the modified Mann-Kendall trend test with a 95% confidence interval (p < 0.05). In Bangladesh, 23% of the rice area shows an increasing trend in boro rice yield, 0.11% shows a decreasing trend, and 76.51% shows no trend.

Given that this is the first attempt to estimate boro rice yield at a 1,000-meter spatial resolution over two decades in Bangladesh, the estimated mid-season boro rice yield estimates are scalable across space and time, offering significant potential for strengthening food security management in Bangladesh. Furthermore, the proposed workflow can be easily applied to estimate rice yields in other regions worldwide.

Keywords: Boro rice yield, MODIS, Random Forest."

3. Introduction section – weak rational justification. How this study can contribute in the literature? What is the hypothesis? These are not clear. There are some related works that used Random Forest model, for example, Ghose et al. 2021.

We have cited the article suggested and the revised sentence now reads in Lines 56-57” Ghose et al. (2021) food that an increase in temperature by 1°C and rainfall by 1% results in a decrease in Aman rice yields by 33.59% and 3.37%, respectively.”

The hypothesis is given in lines 47 - 65 “In recent years, Bangladesh has experienced adverse effects on rice yields due to climate change, characterized by increased extreme weather events, such as droughts and floods during critical phases of rice cultivation (9). This impact is attributed to the strong dependence of rice yields on weather conditions throughout the cultivation stages (10). Moreover, Bangladesh's high vulnerability to climate change, reflected in its seventh-place ranking on the Global Climate Risk Index, raises significant concerns, especially in agriculture and rice production (11). Moreover, Sarker et al., predicted that between 2005 and 2050, rice production in Bangladesh will decline by an average of 7.4% every year due to climate change (12). Likewise, the National Adaptation Plan of Bangladesh (2030-2050) published by the Ministry of Environment, Forest and Climate Change (MEFCC), Bangladesh (13), along with the findings from Kaur el al., highlighted the looming threat posed to this crucial agricultural sector, particularly rice, by the escalating impacts of climate change (14). Ghose et al. (2021) recently revealed that an increase in temperature by 1°C and rainfall by 1% results in a decrease in Aman rice yields by 33.59% and 3.37%, respectively.

The country's vulnerability to these climatic challenges poses a potential threat to food security. It heightens Bangladesh's susceptibility, impacting livelihoods, economic stability, and food security at both national and global levels. This will also pose a significant obstacle to the United Nations Sustainable Development Goals (SDGs), prioritizing achieving zero hunger and promoting sustainable agriculture by 2030 (12). To manage food security, it is therefore important to accurately estimate rice yields, production, and trends, which could also significantly contribute to efforts to lessen climate-related risks to rice production in Bangladesh. Hence, accurate and timely statistics on rice yields, and long-term trends could assist the government of Bangladesh in managing food security and contribute to efforts to lessen climate-related risks to rice production in Bangladesh.”

Lines 82-97 Similar remote sensing methods are utilized in Bangladesh to estimate rice yield, employing spectral bands and vegetation indices from optical and SAR images (37–39). These methods also include regression-based approaches (40), ML models (41), and deep learning (DL) models (42). However, these methods are primarily utilized for research purposes and need to be integrated into official reporting procedures. In addition, the rice yield maps developed from these studies need to be updated, and their coverage is limited to particular geographic regions within the country. Moreover, the rice yield estimation is often conducted at a coarse scale, typically encompassing districts, divisions, or the entire nation, thus needing more spatial and temporal precision. Currently, annual boro rice yield statistics are available from inventory data at the district level; however, no consistent yearly boro rice yield map is available for Bangladesh at a subdistrict spatial scale, such as a 1,000-meter spatial resolution. Due to this limitation, rice trend analyses conducted in other studies (9,43) are typically done at district and national scales, rendering them less useful for planning and implementing policies at the micro-scale (sub-district level).

This study aimed to evaluate the potential of MODIS data and a random forest (RF) ML-based method to develop a rice yield model for estimating boro rice production in Bangladesh from 2021 to 2022 at a spatial resolution of 1,000 meters. Specific objectives were to 1) evaluate RF models for estimating rice yield in Bangladesh, 2) report the accuracy and uncertainty of boro rice yield estimation by RF models, and 3) understand the spatial and temporal trends of rice yield in Bangladesh from 2002 to 2021.

How this study can contribute in the literature?

We clarify that the higher resolution (1000m) boro rice yield estimates compared to existing district-level yield estimates are the most important contribution of this study to the existing body of literature.

 Line 84 - 97 “ However, these methods are primarily utilized for research purposes and need to be integrated into official reporting procedures. In addition, the rice yield maps developed from these studies need to be updated, and their coverage is limited to particular geographic regions within the country. Moreover, the rice yield estimation is often conducted at a coarse scale, typically encompassing districts, divisions, or the entire nation, thus needing more spatial and temporal precision. Currently, annual boro rice yield statistics are available from inventory data at the district level; however, no consistent yearly boro rice yield map is available for Bangladesh at a subdistrict spatial scale, such as a 1,000-meter spatial resolution. Due to this limitation, rice trend analyses conducted in other studies (9,42) are typically done at district and national scales, rendering them less useful for planning and implementing policies at the micro-scale (sub-district level).”

4. In section 2.2, main weakness of the paper is data quality control. The crop yield data was collected during the years 2019-2020 (n= 2,946) and 2020-2021 (n=1845), specifically focusing on five districts (Dinajpur, Rajshahi, Khulna, Jashore and Rangpur) within Bangladesh. Why only five districts were considered for rice yield analysis for this paper?

We used district-level reported rice yield data from all districts in Bangladesh from 2006 to 2021 to train and validate our model. Bangladesh Bureau of Statistics (BBS) reports this dataset based on manual surveys. Additionally, we used high-resolution crop yield data from five districts (Dinajpur, Rajshahi, Khulna, Jashore, and Rangpur) within Bangladesh, obtained by manually harvesting the crop and measuring yield directly and scientifically within farmers’ fields. These districts were selected because we had staff and resources available to undertake an assessment of this scope. Unfortunately, other districts were out of scope for grant funds to permit additional crop yield estimates. Collecting crop yield data from the field is time-consuming, resource-intensive, and costly, making it challenging to gather data from all districts.

It is important to note that this crop-cut yield data serves as a secondary source of validation, while the primary source remains the district-level yield data collected between 2006 and 2021 from all districts of Bangladesh. Additionally, other studies also use district-level yield data for training and validation and conduct field surveys in specific districts only to validate satellite-based estimates with measured ground yields(1–3).

Lines 140 to 144 in the revised manuscript read “The crop yield data was collected during the years 2019-2020 (n= 2,946) and 2020-2021 (n=1845), specifically focusing on five districts (Dinajpur, Rajshahi, Khulna, Jashore and Rangpur) within Bangladesh (Fig 1b). These districts were chosen due to the availability of qualified scientific staff in these districts that could facilitate an ambitious yield measurement program from farmers’ fields. Unfortunately, other districts were out of scope for grant funds to permit additional crop yield estimates..”

Line 354-358 “In this study, boro rice crop yield data was collected from five districts in Bangladesh—Dinajpur, Rajshahi, Khulna, Jashore, and Rangpur (Fig. 1b)—due 

---

## [Decision Letter · Decision Letter 1]

22 Aug 2024

Advancing Food Security: Rice Yield Estimation Framework using Time-Series Optical Data & Machine Learning

PONE-D-24-12288R1

Dear Dr. Tiwari

We’re pleased to inform you that your manuscript has been judged scientifically suitable for publication and will be formally accepted for publication once it meets all outstanding technical requirements.

Kind regards,

Salim Heddam

Academic Editor

PLOS ONE

Reviewer 1#:All comments have been addressed

Reviewer 2#:The authors have addressed all teh queries. Now it can be accepted for publication. paper has improved a lot.

Reviewers' comments:

Reviewer's Responses to Questions

**Comments to the Author**

1. If the authors have adequately addressed your comments raised in a previous round of review and you feel that this manuscript is now acceptable for publication, you may indicate that here to bypass the “Comments to the Author” section, enter your conflict of interest statement in the “Confidential to Editor” section, and submit your "Accept" recommendation.

Reviewer #1: All comments have been addressed

Reviewer #2: All comments have been addressed

2. Is the manuscript technically sound, and do the data support the conclusions?

Reviewer #1: Yes

Reviewer #2: Yes

3. Has the statistical analysis been performed appropriately and rigorously? 

Reviewer #1: Yes

Reviewer #2: Yes

4. Have the authors made all data underlying the findings in their manuscript fully available?

Reviewer #1: Yes

Reviewer #2: Yes

5. Is the manuscript presented in an intelligible fashion and written in standard English?

Reviewer #1: Yes

Reviewer #2: Yes

6. Review Comments to the Author

Reviewer #1: (No Response)

Reviewer #2: The authors have addressed all teh queries. Now it can be accepted for publication. paper has improved a lot.

7. PLOS authors have the option to publish the peer review history of their article (what does this mean?). If published, this will include your full peer review and any attached files.

Reviewer #1: **Yes: **Md Abdur Rouf Sarkar, Bangladesh Rice Research Institute

Reviewer #2: No

---

## [Editor Report · Acceptance letter]

26 Sep 2024

PONE-D-24-12288R1 

PLOS ONE

Dear Dr. Tiwari, 

I'm pleased to inform you that your manuscript has been deemed suitable for publication in PLOS ONE. Congratulations! Your manuscript is now being handed over to our production team.

Kind regards, 

on behalf of

Dr. Salim Heddam 

Academic Editor

PLOS ONE